# Experimental Investigation of Erosion Characteristics of Fine-Grained Cohesive Sediments

**Bommanna Gounder Krishnappan** [1],*, **Mike Stone** [2], **Steven J. Granger** [3], **Hari Ram Upadhayay** [3], **Qiang Tang** [3], **Yusheng Zhang** [3] **and Adrian L. Collins** [3]

[1]    National Water Research Institute, Burlington, ON L7R 4A6, Canada
[2]    Department of Geography and Environmental Management, University of Waterloo, Waterloo, ON N2L 3G1, Canada; mstone@uwaterloo.ca
[3]    Sustainable Agriculture Sciences Department, Rothamsted Research, North Wyke, Okehampton, Devon EX20 2SB, UK; steve.granger@rothamsted.ac.uk (S.J.G.); hari.upadhayay@rothamsted.ac.uk (H.R.U.); qiangtang@imde.ac.cn (Q.T.); yusheng.zhang@rothamsted.ac.uk (Y.Z.); adrian.collins@rothamsted.ac.uk (A.L.C.)
*    Correspondence: krishnappan@sympatico.ca

**Abstract:** In this short communication, the erosion process of the fine, cohesive sediment collected from the upper River Taw in South West England was studied in a rotating annular flume located in the National Water Research Institute in Burlington, Ontario, Canada. This study is part of a research project that is underway to model the transport of fine sediment and the associated nutrients in that river system. The erosion experimental data show that the critical shear stress for erosion of the upper River Taw sediment is about 0.09 Pa and it did not depend on the age of sediment deposit. The eroded sediment was transported in a flocculated form and the agent of flocculation for the upper River Taw sediment may be due to the presence of fibrils from microorganisms and organic material in the system. The experimental data were analysed using a curve fitting approach of Krone and a mathematical model of cohesive sediment transport in rotating circular flumes developed by Krishnappan. The modelled and measured data were in good agreement. An evaluation of the physical significance of Krone's fitting coefficients is presented. Variability of the fitting coefficients as a function of bed shear stress and age of sediment deposit indicate the key role these two factors play in the erosion process of fluvial cohesive sediment.

**Keywords:** erosion; cohesive sediments; rotating circular flume; mathematical modelling; fitting coefficients; sediment deposition; flocculation; bed shear stress; consolidation

## 1. Introduction

Fine-grained cohesive sediment plays an important role in the transportation of pollutants and it is a key driver of water quality degradation in rivers. Rigorous quantification of cohesive sediment transport processes is fundamental for predicting sediment and associated contaminant transport in aquatic systems (Horowitz and Elrick [1]; Luoma and Rainbow [2]). The development of reliable numerical models to simulate cohesive sediment transport dynamics requires an accurate description of fundamental sediment transport processes such as erosion, deposition, and transport of solids in suspension (Grabowski et al. [3]). Factors affecting the erosion characteristics of cohesive sediments include the rate of bed shear (Amos et al. [4]), the degree of consolidation/age of deposit (Lick and McNeil [5]), bio-stabilisation effects by microorganisms (Friend et al. [6]) and the initial conditions that created the deposit (Lau et al. [7]). In a re-examination of the sediment erosion data from flume studies conducted by Roberts et al. [8] and Zreik et al. [9], Krone [10] demonstrated the importance of the

sediment bed matrix structure on erosion characteristics of fine sediment deposits. Factors controlling the deposition characteristics of cohesive sediment include the phyco-chemical properties of sediment water mixture, the concentration of organic matter including microorganisms in the water column and the turbulence characteristics of the flow field which in turn, can cause the suspended sediment particles to flocculate and change their porosity, density and settling characteristics. At the present state of knowledge, numerical models of cohesive sediment transport rely mainly on laboratory experiments using specialised flumes such as a rotating annular flume for the determination of transport parameters that include the critical shear stresses for erosion and deposition, erosion and deposition rates and properties of sediment flocs for site-specific sediments.

A research project is underway in the upper River Taw in South West England to model the transport of fine-grained cohesive sediments. The upper River Taw has been instrumented as a landscape scale observatory for exploring the interactions between climate, land use, farm management and water quality in conjunction with a larger strategic programme exploring pathways for improving the sustainability of agriculture. The upper River Taw drains both organic-rich peaty and podzolic upland moorland soils near its source and clayey gley and brown earth soils in the more intensive agricultural areas (beef and sheep grazing, cereals, maize, oil seed rape) on the more intensively farmed lowland adjacent to the moor. Sediment stress in the study area has been highlighted as a critical factor impacting on lithophilous fish species dependent upon clean bed gravels for their early life stages including the incubation of progeny in redds cut into bed gravels.

As part of the research project on modelling cohesive sediment transport, a survey of 18 cross-sections located ~0.5 km apart was conducted in the upper River Taw (Figure 1) during the summer low flows in 2018. The bank-full width of the study river is ~10 m and the average depth is ~1 m. The riverbed is armoured with large stones including pebbles and cobbles. Bedrock outcrops are also observed. Although the matrix bed material in the study reach is very coarse, fine-grained sediment is present in the river channel bed because of the entrapment process, which retains fine material in the lee of large rocks or directly within the gravel bed matrix as interstitial fines. Fine-grained sediment deposits are also present in recirculation eddy zones along the edges of the river channel where the bed shear stresses of the flow field are low. Representative samples of fine-grained sediment from the study reach were collected using a conventional bed sediment remobilisation technique (Lambert and Walling [11]; Duerdoth et al. [12]; Naden et al. [13]) at all 18 cross sections and 18 one L bottles of sediment slurry were then shipped to Canada for testing in a rotating annular flume located in the National Water Research Institute in Burlington, Ontario, Canada.

In this short communication, preliminary results from this research project on cohesive sediment transport are presented. Specifically, a component of the cohesive sediment transport, namely, the erosion process, was investigated by carrying out erosion experiments using a rotating annular flume. A novel approach was used to analyse the erosion data. The approach is based on a methodology proposed by Krone [10] and a mathematical model of cohesive sediment transport in rotating annular flumes developed by Krishnappan [14].

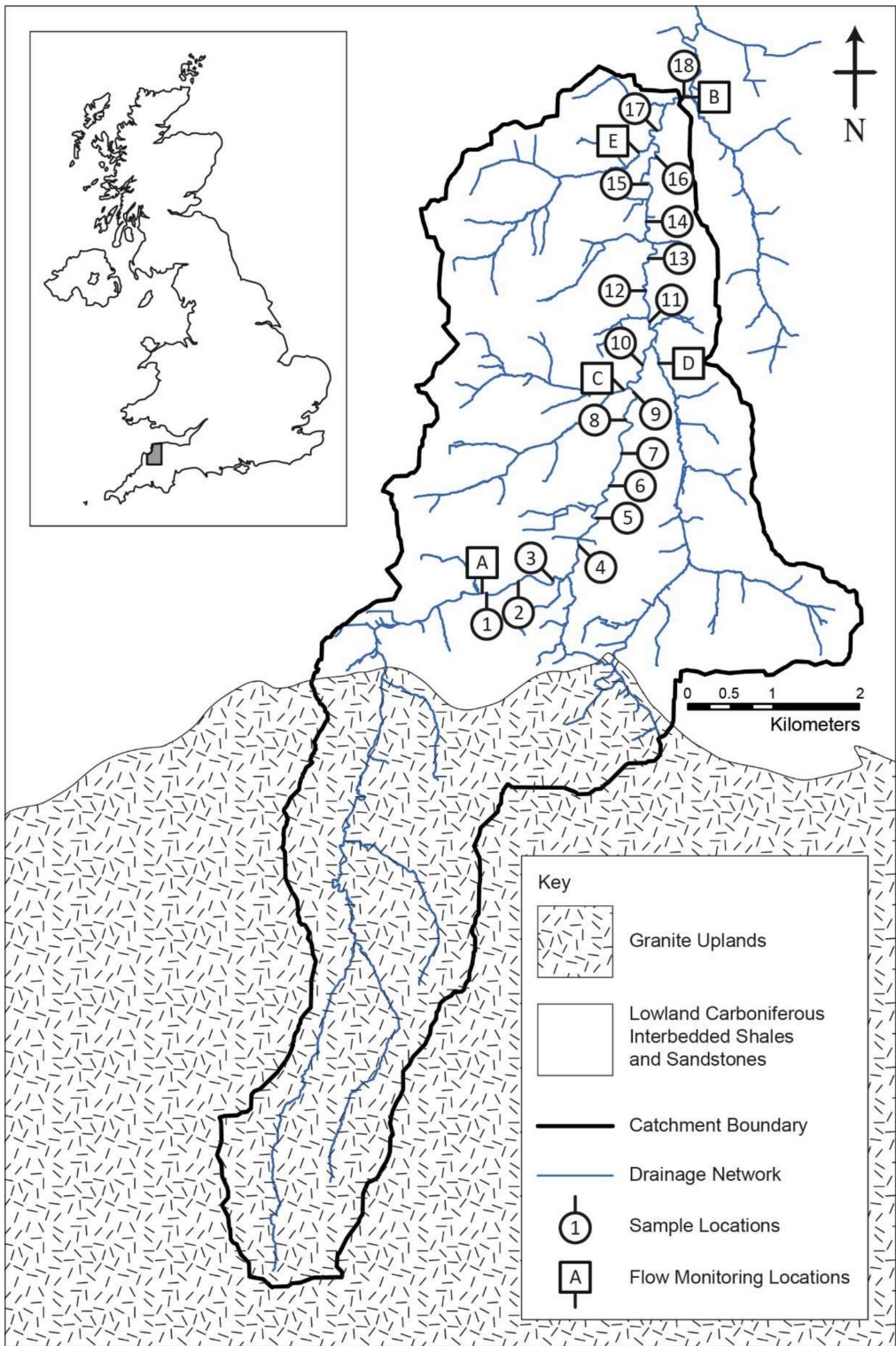

**Figure 1.** Map of the upper River Taw study catchment, showing location in south west England, channel bed cohesive sediment sampling locations and flow gauging stations.

## 2. Material and Methods

### 2.1. Rotating Annular Flume

The rotating circular flume used in this study was 5.0 m in mean diameter, 30 cm wide and 30 cm deep and it rests on a rotating platform which was 7 m in diameter. An annular lid fitted inside the

flume with close tolerance (about 1 mm gap all around) and it rotated in the opposite direction to the flume's rotation. The annular lid maintained contact with the water surface within the flume during experiments. The generated flow fields in such assemblies were nearly two dimensional with near constant bed shear stress across the channel and with minimum secondary circulation in the transverse direction (Petersen and Krishnappan [15]). The flow field in this rotating annular flume assembly was computed using the 3D hydrodynamic flow model, PHOENICS (Rolston and Spalding [16]). The bed shear stress computed by the model was verified using direct measurements of bed shear stress using a preston tube (Krishnappan and Engel [17]). The main advantage of rotating flumes over linear flumes is that detrimental effects of the pump and the pipe system on the floc structure of the cohesive sediment were avoided, thereby permitting reliable studies on floc behaviour to be conducted. A complete description of the flume can be found in Krishnappan [18].

The sediment samples shipped from the UK were stored in a cold room until the start of the experiments. A large composite sediment sample was prepared by combining all 18 bottles of samples and wet sieved using a 200 mesh sieve before being added to the flume. The erosion experiments were conducted by using the standard procedure which involves operating the flume at high speed initially to suspend the sediment and then lowering the flume speed gradually to allow the suspended sediment to deposit and form a uniform bed. The bed was then allowed to age for three different time periods (herein referred to as age of deposit), namely, 22 h, 38 h and 160 h. During an erosion test, bed shear stresses were increased over time in steps as a stair-case function. The shear stress steps used were 0.06, 0.09, 0.12, 0.17, 0.27 and 0.33 Pa, with each maintained for a period of one hour. In each shear stress step, sediment samples were collected every 10 min to determine the concentration of the eroded sediment as a function of time. When the concentration reached a steady-state value, the flume speed was increased to the next level. Whenever there was sufficient sediment suspended in the water column, sediment samples were collected for size analysis using a LISST (Laser In-situ Scattering and Transmissometry 100X; Sequoia Scientific, Bellevue, WA, USA) particle size analyser and an image analysis system. This procedure was repeated until the maximum permissible flume speed was reached.

### 2.2. Methodology of Krone

Krone [10] developed his methodology when he analysed erosion data reported by Zreik et al. [9] who studied the erosion behaviour of cohesive sediments from Boston Harbour using a rotating circular flume. Measured sediment concentration data from Zreik et al. [9] showed that the concentration of eroded sediment increased rapidly at the beginning of each applied shear stress step change but then levelled off with time before the next incremental step in shear stress.

Krone [10] fitted the following equation to describe the variation in sediment concentration during a shear stress step:

$$C = \frac{\frac{1}{c_1}t}{\frac{c_0}{c_1}t + 1} \tag{1}$$

where $C$ = incremental concentration of eroded sediment during a shear stress step; t = time after the shear stress change; $c_0$ and $c_1$ = constants. This equation fitted the experimental data of Zreik et al. [9] very well and, importantly, the constants $c_0$ and $c_1$ have physical meaning. For example, when t tends to infinity, the ratio $1/c_0$ takes on the value of the concentration near the end of the shear stress step in question. When t = 0, the ratio $1/c_1$ assumes the value of the erosion rate, i.e., $dC/dt = 1/c_1$ at $t = 0$. The constants $c_0$ and $c_1$ were evaluated by knowing sediment concentrations at two different times (near the beginning and near the end of the shear stress step) using the following relationships:

$$c_1 = \left(\frac{1}{C_T} - \frac{1}{C_L}\right)T \qquad \text{and } c_0 = \frac{1}{C_L} \tag{2}$$

where $C_T$ = concentration near the beginning of the shear stress step, $C_L$ = limit concentration at the end of the shear stress step and $T$ is the elapsed time from the beginning of the shear stress step till the time when $C_T$ is specified. Knowing $c_0$ and $c_1$, Equation (1) can be used to calculate the concentration of the eroded sediment for the entire duration of each shear stress step. In addition, Equation (1) is used to derive an erosional rate function that was used in the model of cohesive sediment transport developed by Krishnappan [2]. The erosion rate of sediment can be expressed as follows:

$$E = h\frac{dC}{dt} \tag{3}$$

where $E$ is the erosion rate in gm/m$^2$s, h is the depth of water in the flume in metres and $dC/dt$ is the concentration gradient which can be evaluated from Equation (1). Substituting the expression of $dC/dt$ in Equation (3), the erosion rate function can be derived as:

$$E = h\frac{(1/c_1)}{\left(\frac{c_0}{c_1}t + 1\right)^2} \tag{4}$$

*2.3. Mathematical Model of Cohesive Sediment Transport Developed by Krishnappan*

A mathematical model of cohesive sediment transport in the rotating circular flume (called the FLUME model) was used to simulate the erosion process of the sampled upper Taw River sediment. The FLUME model incorporates the erosion rate function of Krone [10] i.e., Equation (4). A full description of the model can be found in Krishnappan [14]. Here, a summary of the salient features of the model is outlined for the sake of completeness. The FLUME model treats the motion of sediment in the rotating flume in two stages: a transport/settling stage and a flocculation stage. Some salient features of these two fundamental stages of sediment transport are briefly discussed below:

2.3.1. Settling Stage

The governing equation for the settling stage was obtained from mass balance considerations. Assuming that the flow in the rotating flume is uniform in the longitudinal and tangential directions, the one-dimensional mass balance equation as shown below was adopted:

$$\frac{\partial C_k}{\partial t} + w_k\frac{\partial C_k}{\partial z} = \frac{\partial}{\partial z}\left(\Gamma\frac{\partial C_k}{\partial z}\right) + S \tag{5}$$

where, $C_k$ is the concentration of sediment in size fraction $k$, $w_k$ is the settling velocity of that fraction and $\Gamma$ is the dispersion coefficient in the vertical direction. The symbol $t$ represents the time axis and $z$ represents the coordinate axis in the vertical direction. $S$ is the source/sink term. The boundary conditions used for the settling stage are:

At the free surface:

$$-w_kC_k - \Gamma\frac{\partial C_k}{\partial z} = 0 \tag{6}$$

At the bed:

$$-w_kC_k - \Gamma\frac{\partial C_k}{\partial z} = q_e + q_d + q_{entrap} \tag{7}$$

The free surface boundary condition expresses the balance between the settling flux and the dispersive flux so that there is no external input of sediment at this boundary. At the bed surface, it is assumed that the settling and dispersive fluxes are balanced by the net amount of sediment exchanged at the sediment-water interface. The sediment exchange at the sediment-water interface can occur when: (1) sediment is eroded from the bed and entrained into the water column ($q_e$) (2) sediment settles to the bed and stays on the bed as the deposited sediment ($q_d$) (3) sediment settles to the bed and a

portion of the deposited sediment can ingress into the interstitial pores of the gravel bed and become unavailable for further erosion and entrainment ($q_{entrap}$).

The first two components of the sediment exchange at the sediment-water interface have been studied extensively in the literature, including by Partheniades [19], Mehta and Partheniades [20], Parchure [21], Lick [22], Krone [10], Krishnappan [23], Krone [24], Mehta and Partheniades [20], and Lick [22]. In this study, the approaches proposed by Krone [10] and Krone [24] for erosion and deposition respectively have been adopted. Accordingly, the erosion flux was calculated as:

$$q_e = E \tag{8}$$

where the erosion rate, $E$ is given by Equation (4) and the deposition flux was calculated as:

$$q_d = P w_k C_{kb} \tag{9}$$

where $P$ is a probability parameter which gives a measure of the probability that a sediment particle settling to the bed, stays at the bed. $C_{kb}$ is the near-bed concentration of the sediment fraction $k$. Krone [24] proposed a relationship for $P$ as:

$$P = \left(1 - \frac{\tau}{\tau_{crd}}\right) \tag{10}$$

where $\tau$ is the bed shear stress and $\tau_{crd}$ is the critical shear stress for deposition, which is defined as the bed shear stress above which none of the initially suspended sediment would deposit.

The third component, namely, the entrapment component, is assumed to be directly proportional to the settling flux near the bed and is expressed as:

$$q_{entrap} = \alpha w_k C_{kb} \tag{11}$$

where $\alpha$ is the proportionality constant and is termed the entrapment coefficient. The entrapment coefficient is expected to be a function of porosity of the gravel substrate, the thickness of the gravel bed layer and the permeability of the substrate. Further research is needed to quantify this term as a function of the bulk properties governing the entrapment process.

### 2.3.2. Flocculation Stage

The flocculation stage was modelled using the coagulation equation (Fuchs [25]) shown below:

$$\frac{\partial N(i,j)}{\partial t} = -\beta N(i,t) \sum_{i-1}^{\infty} K(i,j)N(j,t) + \frac{1}{2}\beta \sum_{i-1}^{\infty} K(i-j,j)N(i-j,j)N(j,t) \tag{12}$$

where $N(j,t)$ and $N(j,t)$ are number concentrations of particle size classes $i$ and $j$, respectively. $K(i,j)$ is the collision frequency function, which is a measure of the probability that a particle of size $i$ collides with a particle of size $j$ in unit time and $\beta$ is the cohesion factor, which defines the probability that a pair of collided particles will coalesce and form a new floc. The $\beta$ term accounts for the effects of cohesion properties such as the electrochemical properties of the sediment-water mixture and the effects of polymers secreted by microorganisms. In this study, is treated as an empirical parameter and was determined through the calibration of the model using experimental data from the flume.

The first term on the right side of Equation (12) gives the reduction of particles in size class $i$ because of flocculation of these particles in size class $i$ with all other size classes. The second term gives the generation of new particles in the size class $i$ because of flocculation of particles in smaller size classes. In this process, the mass of particles is assumed to be conserved. Equation (12) was simplified by considering the particle sizes in discrete size classes where the continuous particle size space is considered in discrete size ranges. Each range is considered as a bin containing particles of a certain

size range. For example, $r_i$ is the geometric mean radius of particles in bin 1. The particle ranges were selected in such a way that the mean volume of particles in bin $i$ is twice that of the preceding bin.

Under this scheme, when particles of bin $i$ flocculate with particles in bin $j$ ($j < i$), the newly formed particles will fit into bins $i$ and $i + 1$. The proportion of particles going into these bins can be calculated by considering the mass balance of particles before and after flocculation. With this simplification, the coagulation equation can be expressed in discrete form as follows:

$$\frac{\Delta N_i}{\Delta t} = -\sum \beta K(r_i, r_j) \, N_i N_j + \sum f_{i,j} \beta K(r_i, r_j) N_i N_j + \sum (1 - f_{i,j}) \beta K(r_i, r_j) N_{i-1} N_j \tag{13}$$

where $f_{i,j}$ is the allocation function given by:

$$f_{i,j} = (\rho_{s,j} V_i + \rho_{s,j} V_j - \rho_{s,j+1} V_{i+1}) / (\rho_{s,i} V_i - \rho_{s,j+1} V_{i+1}) \tag{14}$$

where, $\rho_s$ denotes the density of the flocs and $V$ is the volume of the flocs. The density of the flocs is dependent on floc size and, here, an empirical relationship proposed by Lau and Krishnappan [26] was adopted in this study. The form of the relationship is as follows:

$$\rho_s - \rho = (\rho_p - \rho) \exp(-bD^c) \tag{15}$$

where $\rho_s$, $\rho$ and $\rho_p$ are densities of flocs, water and parent material forming the flocs, respectively. $D$ is the diameter of the floc and b and c are empirical coefficients to be determined through model calibration. The settling velocity of flocs, needed for the settling stage of the sediment particle motion, was calculated using the above density relation and Stoke's Law. The resulting expression is as follows:

$$w_k = \left(\frac{1.65}{18}\right)\left(\frac{gD_k^2}{v}\right)\exp(-bD^c) \tag{16}$$

where $g$ stands for acceleration due to gravity and $v$ stands for the kinematic viscosity of water.

The collision frequency function $K(i, j)$ takes different functional forms depending on the collision mechanisms that bring the particles to close proximity. The mechanisms considered in this study include (1) Brownian motion ($K_b$); (2) turbulent fluid shear ($K_{sh}$); (3) inertia of particles in turbulent flow ($K_I$), and; (4) differential settling of flocs ($K_{ds}$). An effective collision frequency function, $K_{ef}$, was calculated in terms of the individual collision functions as follows:

$$K_{ef} = K_b + \sqrt{\left(K_{sh}^2 + K_I^2 + K_{ds}^2\right)} \tag{17}$$

The collision frequency functions for the individual collision mechanisms can be found in Valioulis and List [27].

The break-up of flocs due to turbulent fluctuations of the flow was modelled using a scheme proposed by Tambo and Watanabe [28]. According to this scheme, a "collision-agglomeration" function was introduced as a multiplier to the collision frequency function $K_{ef}$. This produced an effective collision frequency that resulted in an optimum floc size distribution for a given level of turbulence. The function proposed by Tambo and Watanabe [28] is as follows:

$$\alpha_R = \alpha_0 \left(1 - \frac{R}{R_m + 1}\right)^n \tag{18}$$

where $R$ is the number of primary particles contained in a floc and $R_m$ is the number of particles contained in the biggest floc for the given level of turbulence. The parameters $\alpha_0$ and $n$ assume values of 1/3 and 6, respectively. The flow field and the turbulence characteristics in the rotating circular flume needed for the FLUME model were predicted using PHOENICS (Rosten and Spalding [16]).

The FLUME model was used in this study to simulate the erosion process, even though the model is capable of predicting the total transport of cohesive sediment in rotating circular flumes including the deposition and the entrapment processes. The FLUME model, therefore, can be used to determine all parameters governing the transport of cohesive sediments by carrying out experiments using a rotating circular flume and then use these parameters in models that can predict the transport of cohesive sediment under field conditions.

## 3. Results and Discussion

The results of erosion tests from all three runs carried out for the present study are presented in Figure 2. The erosion data indicate that the sediment deposit was completely stable until the shear stress reached a value of 0.09 Pa, which can be considered as the critical shear stress for the erosion of the surficial layer of the sediment deposit. The data presented also show that six out of the seven shear stress steps at each age of deposit produced erosion of the riverbed sediment. At each of the six shear stress steps when erosion of the sediment bed occurred, the eroded sediment concentration increased suddenly after the shear stress change, suggesting that the sediment erosion follows the pattern of "bulk" erosion. As the sediment erosion continues, the erosion rate decreases due to the increasing strength of the bed with bed depth. Erosion during the later stages of the shear stress step is defined as the "surface" erosion when the removal of the material from the bed is particle by particle and the concentration in the water column approaches a steady state value. Additionally, the influence of the age of deposit is evident for the lower shear stress steps, since the concentration of the eroded sediment decreased as the age of deposit increased (Figure 2). For higher shear stress steps (i.e., for 0.27 and 0.33 Pa), the influence of age of deposit is not as pronounced as for the lower shear stress steps.

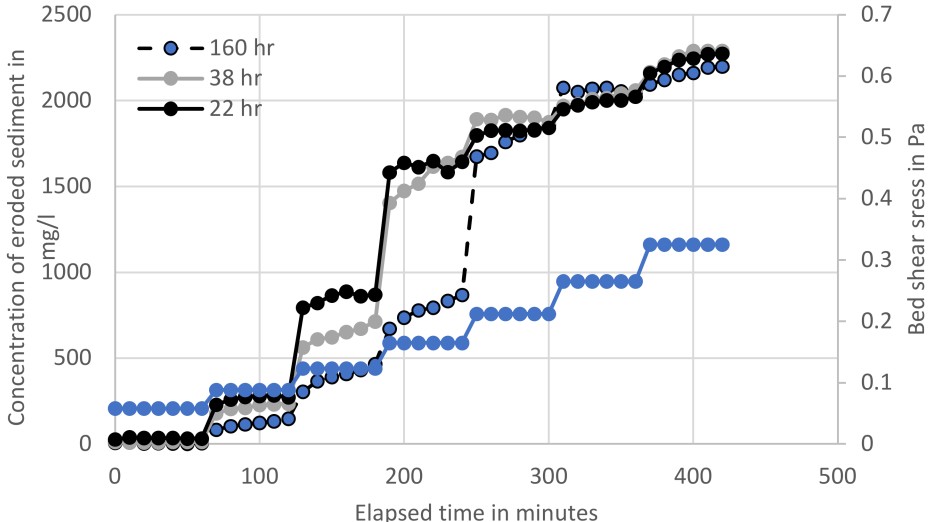

**Figure 2.** Results from the erosion tests.

The size distribution of the eroded sediment measured using a LISST and the image analysis system indicate that the sediment is flocculated. Figure 3 shows the size distribution data measured using LISST for the shear stress step of 0.33 Pa and where the age of deposit is 160 h.

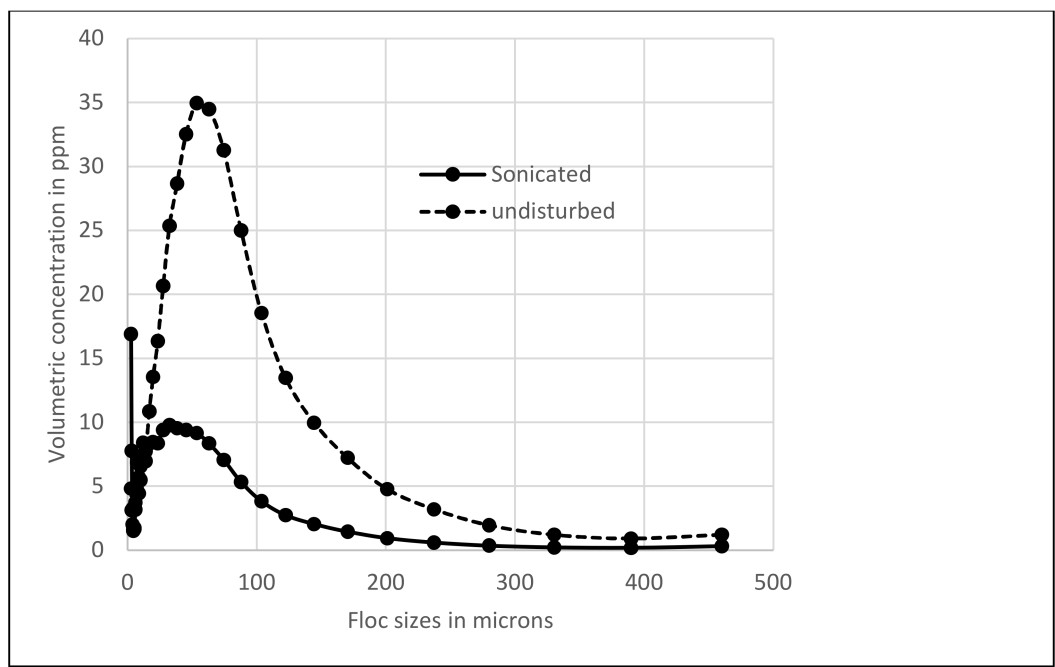

**Figure 3.** Size distribution of eroded sediment at the shear stress step of 0.33 Pa and age of deposit equal to 160 h.

The undisturbed and sonicated size distribution of eroded sediment for one set of experimental conditions (0.33Pa and deposit age of 160 h) are presented in Figure 3. The dotted line represents the size distribution of the eroded sediment as determined by the LISST for the undisturbed sample, whereas the solid line represents the distribution for the sonicated sample. Sonication breaks up all the flocs that are present in the sample. From these two distributions, we can conclude that the eroded sediment in the water column is in a flocculated state and the sediment exhibits cohesive tendencies. The change in the predominant size of the flocculated sediment from ~50 μm to ~25 μm after sonication highlights how flocculation can influence particle morphology and, by extension, volumetric concentration (Figure 3). A particle size of 63 microns is considered in the literature as the division between cohesive and non-cohesive sediments.

A photomicrograph of the eroded sediment sample collected for the image analysis is shown in Figure 4.

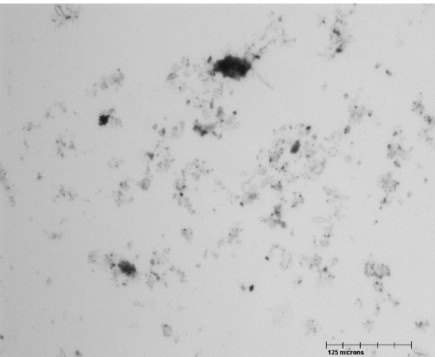

**Figure 4.** A photomicrograph of the eroded sediment for shear stress step of 0.33 Pa and age of deposit equal to 160 h.

This figure provides visual evidence of flocculation of the particles and also shows that mico-flocs are the building blocks of larger flocs as previously reported by Stone et al. [29]. The particles are interconnected through loose fibril material that might have been secreted by the microorganisms or the

organic material that is present in the system. Therefore, flocculation is enhanced by microorganisms and the organic material and this process will have to be accurately represented in models to simulate sediment transport in the upper River Taw River.

Krone's curve fitting approach was used for the data shown in Figure 2. For each shear stress step where the sediment erosion was present, the constants $c_0$ and $c_1$ were calculated using Equation (2) with T = 10 min. The values of these constants, termed herein as "fitting coefficients" are listed in Table 1.

**Table 1.** Fitting coefficients $c_0$ and $c_1$

| Shear Stress Steps | Age of Deposit 22 h | | Age of Deposit 38 h | | Age of Deposit 160 h | |
|---|---|---|---|---|---|---|
| | $c_0$ (m³/gm) | $c_1$ (m³sec/gm) | $c_0$ (m³/gm) | $c_1$ (m³sec/gm) | $c_0$ (m³/gm) | $c_1$ (m³sec/gm) |
| 0.09 Pa | $4.2 \times 10^{-3}$ | 0.57 | $4.6 \times 10^{-3}$ | 0.88 | $7.1 \times 10^{-3}$ | 3.65 |
| 0.12 Pa | $1.7 \times 10^{-3}$ | 0.15 | $2.1 \times 10^{-3}$ | 0.57 | $3.2 \times 10^{-3}$ | 1.95 |
| 0.17 Pa | $1.3 \times 10^{-3}$ | 0.07 | $1.0 \times 10^{-3}$ | 0.24 | $2.5 \times 10^{-3}$ | 1.43 |
| 0.21 Pa | $5.1 \times 10^{-3}$ | 0.87 | $4.4 \times 10^{-3}$ | 0.18 | $1.0 \times 10^{-3}$ | 0.14 |
| 0.27 Pa | $5.5 \times 10^{-3}$ | 2.22 | $5.4 \times 10^{-3}$ | 3.08 | $4.6 \times 10^{-3}$ | 0.35 |
| 0.33 Pa | $4.0 \times 10^{-3}$ | 2.03 | $4.4 \times 10^{-3}$ | 3.05 | $6.6 \times 10^{-3}$ | 8.57 |

With values of $c_0$ and $c_1$, Equation (1) was applied to all the shear stress steps and the concentration variation as a function of time was calculated. A comparison of the fitted concentration variation with the measured data is presented in Figure 5 for a shear stress step of 0.17 Pa and consolidation time of 38 h as an example. The modelled and measured data are in agreement for all six shear stress steps in all three sediment consolidation time tests.

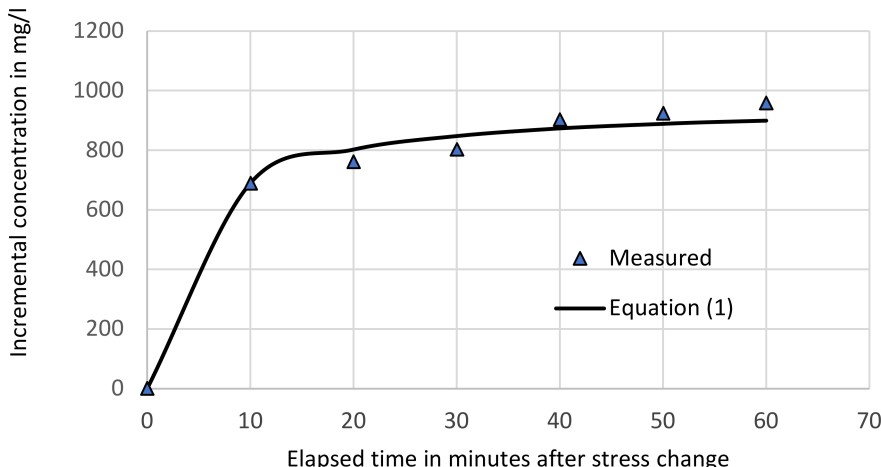

**Figure 5.** Comparison between measured and fitted sediment concentrations using Equation (1) for shear stress step: 0.17 Pa; Age of deposit: 38 h ($r^2$ = 0.993).

The FLUME model was applied to the present erosion experiments to simulate the concentration of the eroded sediment in the water column for all three experiments. The model was applied to each shear stress step using the appropriate parameters, $c_0$ and $c_1$, corresponding to that particular shear stress step and the age of the deposit. The deposition and entrapment fluxes were suppressed for the present simulations. The simulated concentration of the eroded sediment is compared to the measured concentration in Figure 6 and a favourable agreement between the two was observed. The erosion rate function that was derived from the fitting equation (Equation (1)) of Krone [10] works well with the model able to produce sediment concentrations that agree well with the measured data.

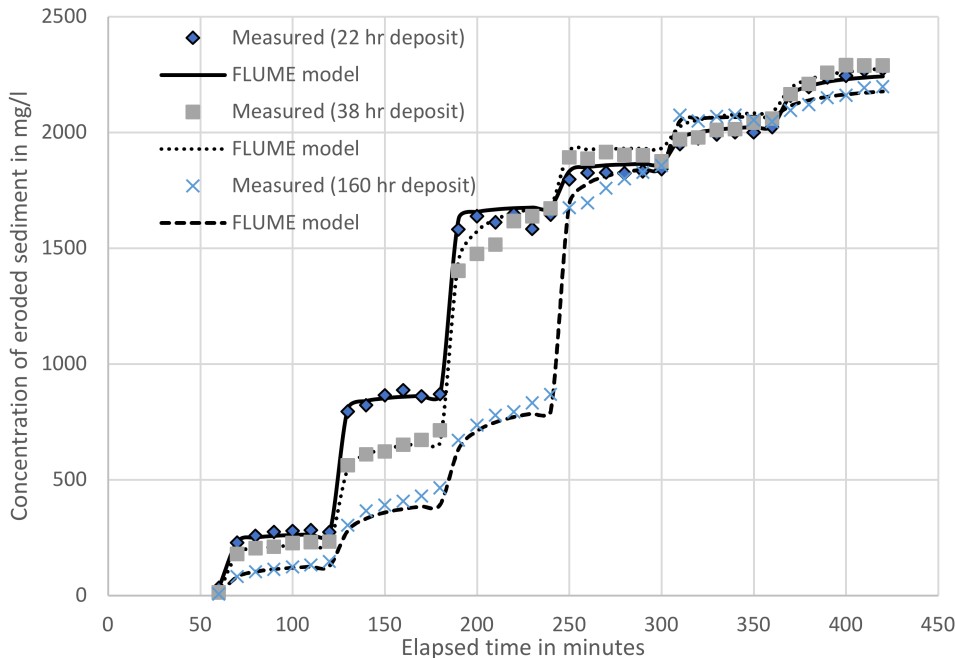

**Figure 6.** Comparison of measured and modelled sediment concentrations predicted by the FLUME model.

The fitting coefficients, $c_0$ and $c_1$, have different values for each shear stress step and age of deposit (Table 1). The variability of these coefficients with each shear stress and age of deposit is shown in Figure 7 which shows that these coefficients vary as a function of both shear stress and age of the deposit.

The coefficients exhibit a complex behaviour with respect to bed shear stress. Both coefficients decrease initially as the shear stress increases, reach minimum values and then increase as the shear stress is increased further. Notably, this behaviour was observed for all three ages of deposit tested. The minimum conditions for the coefficients imply maximum erosion rate and the maximum amount of sediment eroded (implying "bulk" erosion). The minimum condition for the coefficients shifts to the right as the age of deposit increases (Figure 7). Notably, when the age of deposit is 22 h, minimum conditions for both coefficients are at around 0.15 Pa, whereas for the age of deposit of 160 h, the minimum conditions are shifted to 0.21 Pa for $c_0$ and to 0.25 Pa for $c_1$. Accordingly, this suggests that the bed shear strength has increased due to the age of the deposit and higher shear stresses are needed to cause "bulk" erosion. The variability of the fitting coefficients demonstrates the importance of the roles that the bed shear stress and the age of deposit play in determining the erosion behaviour of the upper River Taw sediment. This finding will be useful in developing a modelling framework for predicting cohesive sediment transport in the upper River Taw.

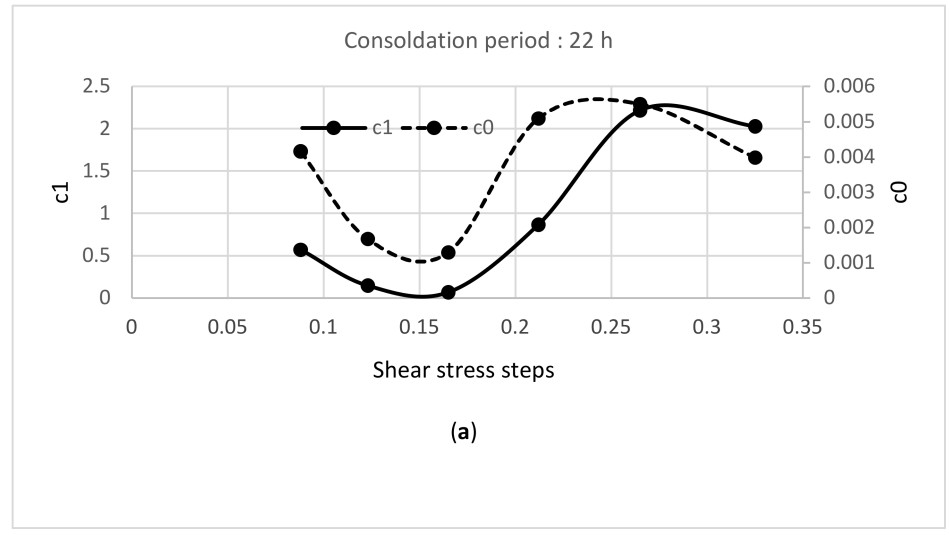

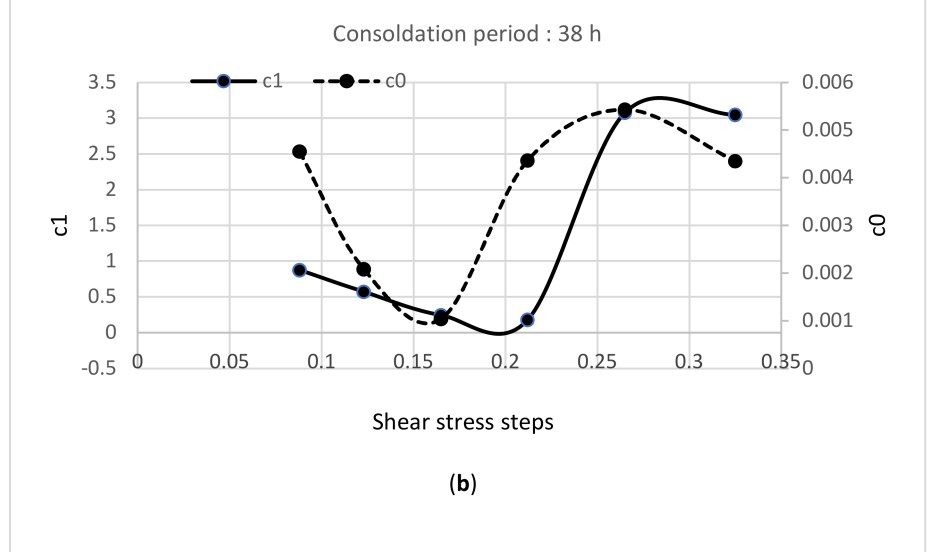

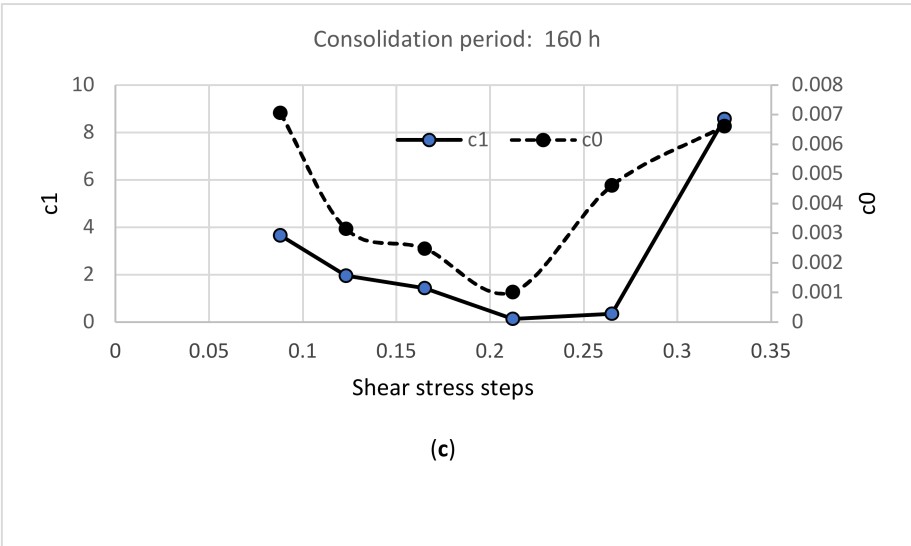

**Figure 7.** The variability of the fitting constants as a function of shear stress and age of deposit (**a**): 22 h; (**b**): 38 h and (**c**): 160 h.

## 4. Conclusions

A better understanding of cohesive sediment transport processes is required to develop models that can simulate scenarios that address both scientific and policy questions. In this short communication, the erosion process of the fine, cohesive sediment collected from the upper River Taw in South West England was studied by conducting erosion experiments using a rotating annular flume located in the National Water Research Institute in Burlington, ON, Canada. The erosion experimental data show that the critical shear stress for erosion of the upper River Taw sediment is about 0.09 Pa and it did not depend on the age of the deposit. The size distribution measurements for the eroded sediment indicates that the eroded sediment was transported in a flocculated form and hence the upper River Taw sediment exhibits cohesive properties. Photomicrographs of the eroded sediment samples obtained by image analysis suggest that the agent of flocculation for the sampled sediment may be due to the presence of fibrils from microorganisms and organic material present in the River Taw system. The experimental data from the erosion experiments were analysed using an approach proposed by Krone [10] and a mathematical model of cohesive sediment transport in rotating annular flumes developed by Krishnappan [14]. The approach of Krone [10] was applied to the experimental data and the fitting coefficients were established as a function of bed shear stress and age of sediment deposit. Using the approach of Krone [1], an erosion rate function was calculated and was used in the FLUME model of Krishnappan [14]. A comparison of the model predictions and the experimental data confirmed that the fine sediment transport model is capable of simulating accurately the erosion experiments in a rotating circular flume. The variability of the fitting coefficients, as a function of bed shear stress and the age of the deposit, was also examined in the present study. Future work will incorporate the experimental results reported herein into a catchment scale cohesive sediment transport modelling framework for exploring the implications of future climate and land management scenarios on sediment transport and behaviour.

**Author Contributions:** Conceptualization, B.G.K., M.S. and A.L.C.; methodology, B.G.K.; Software, B.G.K.; Validation, S.J.G., H.R.U., Q.T. and Y.Z.; formal analysis, B.G.K.; investigation, S.J.G., H.R.U., Q.T. and Y.Z.; resources, S.J.G.; data curation S.J.G., H.R.U., Q.T. and Y.Z.; writing-original draft preparation, B.G.K.; review and editing, M.S. and A.L.C.; visualization, S.J.G.; supervision, M.S. and A.L.C.; project administration, M.S. and A.L.C.; funding acquisition, A.L.C. and M.S. All authors have read and agreed to the published version of the manuscript.

**Funding:** Rothamsted Research received strategic funding from UKRI-BBSRC (UK Research and Innovation—Biotechnology and Biological Sciences Research Council) and the contribution of institute staff to this paper was funded by the Soil Nutrition institute strategic programme project 3 grant (BBS/E/C/000I0330). Funding for the flume experiments was from an NSERC Discovery Grant RGPIN-2020-06963 awarded to M. Stone.

**Acknowledgments:** The authors acknowledge the cooperation of Ian Droppo of Environment and Climate Change Canada and the use of the Rotating Circular Flume for their study. The flume experiments were carried out by Robert Stephens, a retired Research Technician of Environment and Climate Change Canada.

**Conflicts of Interest:** The authors declare no conflict of interest.

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
