# Peer review of "Experimental Investigation of Erosion Characteristics of Fine-Grained Cohesive Sediments"

_water, doi:10.3390/w12051511_

Round 1
Reviewer 1 Report
I have reviewed the contributed manuscript 'Experimental investigation of erosion characteristics of cohesive sediments' and find it needs minor revision prior to acceptance and publication. The manuscript is well organized and written, utilized proper methods, and the conclusions are justified by the data presented. The authors have, indeed, presented a very convincing argument supporting their conclusions. I have only a few comments the authors should consider and address.
In the Materials and Methods section, the details of sediment collection and shipment are lacking. The collection and handling of the sediments between the native locations and testing in the laboratory flume thousands of miles away is critical to the testing outcomes and should be detailed to allow others to duplicate the method.
L120: Please define consolidation period. I suspect it has something to do with sample handling, but I don't have sufficient information to be confident.
L165: the 'alpha' in equation (11) does not match the 'alpha' in subsequent text.
L240: You provide data in Fig. 1 from experiments that you do not discuss. This is confusing. I recommend that you either remove the figure or present a short description of the deposition phase of the experiment. If you are publishing the deposition phase of the experiment as a separate manuscript, then removing Fig. 1 would be the appropriate approach.
L294: The indentation and capitalization of 'S' hould be removed so that it does not confuse the structure of the sentence begun on L292.
This is a very well conceived and presented investigation. Once the authors address the issue that I and other reviewers raise, I don't think it will need further review. You may want to talk with the folks who manage your review engine. It is very inconvenient that reviews cannot simply be copied and pasted from a common word processor or from a PDF file.
Author Response
Reviewer 1 comments and authors' reply. Authors' replies are shown in bold italics.
I have reviewed the contributed manuscript 'Experimental investigation of erosion characteristics of cohesive sediments' and find it needs minor revision prior to acceptance and publication. The manuscript is well organized and written, utilized proper methods, and the conclusions are justified by the data presented. The authors have, indeed, presented a very convincing argument supporting their conclusions. I have only a few comments the authors should consider and address.
Response: Thank you for providing the review and for the very positive comments towards the draft m/s.
In the Materials and Methods section, the details of sediment collection and shipment are lacking. The collection and handling of the sediments between the native locations and testing in the laboratory flume thousands of miles away is critical to the testing outcomes and should be detailed to allow others to duplicate the method.
Response: We agree with the reviewer and addressed the issue in the revised m/s. We have added a new figure (Figure 1) in the revised m/s showing the sampling locations in the upper River Taw study. We have added details of sample collection on lines 81-86 of the revised m/s. We have added sample preparation details on lines 115-117 of the revised m/s.
L120: Please define consolidation period. I suspect it has something to do with sample handling, but I don't have sufficient information to be confident.
Response: We have replaced the phrase "consolidation period" with "age of deposit" to clarify that we allowed the suspended sediment to deposit in the flume to form a sediment bed. This sediment bed was allowed to age a certain amount time that we now refer to as "age of deposit" now. See lines: 119-122. This new term is now used consistently throughout the revised m/s including for example in the revised Table 1 and on line 297.
L165: the 'alpha' in equation (11) does not match the 'alpha' in subsequent text.
Response: This has been fixed. See lines 207 and 208 in the revised m/s.
L240: You provide data in Fig. 1 from experiments that you do not discuss. This is confusing. I recommend that you either remove the figure or present a short description of the deposition phase of the experiment. If you are publishing the deposition phase of the experiment as a separate manuscript, then removing Fig. 1 would be the appropriate approach.
Response: We have removed this figure as suggested.
L294: The indentation and capitalization of 'S' hould be removed so that it does not confuse the structure of the sentence begun on L292.
We have done this.
Reviewer 2 Report
I would like to thank you very much for the opportunity to review the article „Experimental Investigation of Erosion Characteristics of Cohesive Sediments.” I find the article interesting, the subject matter is essential from the point of view of erosion of the bed or river banks, transport of river debris and silting of water reservoirs or backfilling of the river from its mouth.
However, before publication, the article requires a thorough rewriting. Needs additions of omitted descriptions of the material, the test methodology, an indication of a clear research objective, a sign of the originality of the article, and evidence of what new it brings to science.
The title of the article states, 'Cohesive Sediments.' Please explain what speaks for using the term "cohesive." From the point of view of grain size, it is non-cohesive material. In the manuscript, there is no description of this material to confirm this property, perhaps beyond the coagulation (flocculation) property given in the text. It would be appropriate to use the term "fine-grained material" - fine sediment, also used in the text.
The title of the article states, 'Cohesive Sediments.' Please explain what speaks for using the term "cohesive." From the point of view of grain size, it is non-cohesive material. There is no description of this material to confirm this property, perhaps beyond the coagulation property given in the text.
Throughout the article, we find many references to literature. There are nine items, including seven by one author (Krishnappan, B. G., items 2, 9, 11, 15, 16, 23, 26) and two by another author (Collins, A.L., items 18, 19). Seven authors write the article. Therefore, please specify the role and percentage of each author in the report. This is a current practice in many journals.
So many references to literature related to the content of the article raise controversy as to whether the material is original and what brings new to science. Please address my concerns.
Notes to the Abstract. Too general and contains known knowledge of factors influencing erosion process in river beds. Bed shear stress and consolidation, however, are not the only factors, but I understand that boundary conditions to the model need to be generalized. It is also unknown what kind of debris is being considered. They are all alluvial deposits — rolling, sliding, and skipping. There is nothing mentioned about that. There is a lack of knowledge about the particles, granulometry, organic matter content, chemical composition of the material, the influence of agricultural management of adjacent areas, etc. Lack of information about the innovation of research, modeling, and results obtained.
Notes to the Introduction. It contains only a literature review, foreign and own. It follows that the article includes only repeated research and reference to existing theories and models. It consists of a description of the catchment area, land use adjacent to the river, geomorphology. But it is not then analyzed anywhere, e.g., in case of coagulation of silt particles.
Comments on the chapter Materials and Methods. Too general and referring directly to the literature. The chapter needs to be supplemented with a diagram of the test, a photo of the test stand, a way of mapping the in-situ state of the material from the river bed. The reader, and especially the reviewer, does not evaluate the cited literature but this particular article. Usually, the research is performed on many models and repetitions. Why cohesive samples, lack of explanations. There is one study described in the article, is it? Why? The shear stresses used in the test are in the order of 9E-2 Pa. Is it measured with such accuracy or recalculated? Why is this test methodology used? The applied mathematical model (Flume Model) is also used behind the literature (Krone). What's your own?
The settling stage and the Flocculation stage were adapted from the cited literature. Where is the originality of the article?
Notes to the Results and discussion chapter. No presentation of the grain-size curve of the test material. Figure 1 presents only the distribution of the size of individual fractions and their percentage content by volume. This is an interesting presentation in relation to grain size changes before and after the test. In conclusion, the material has been classified as cohesive.
Please give the grain-size curve of the tested material and define it with the indicators based on grain size known from the literature. This will allow us to evaluate the material as one fractional or multi-fractional, show its resistance to filtration phenomena such as suffusion, etc. What does consolidation mean in this case? Consolidation is a process related to the load, the formation of excess water pressure in pores, and its dispersion. It is a process that occurs in a long time. It depends on the grain size of the particles, and the drainage conditions of the loaded layer, one or two vertical directions. What kind of settlement: immediate in undrained conditions, consolidation faze or secondary. What type of sedimentation: immediate, consolidation phase connected with the dissipation of excess pore water pressure or secondary in effective stress. How is it in this case?
What kind of test was performed to determine the grain size distribution? What is the procedure and type of device used? The bed shear stress values used in the analysis are very low. The testing time is short. What does this result from?
The values given in figure 3 do not correlate with those given in table 1. The signature under the figure also. Why? What the authors wanted to show in the drawing. Please explain.
Notes to the chapter Conclusion. I don't know what to write. It's very general now. The chapter can be rewritten after all comments and corrections have been made.
Author Response
Reviewer 2 comments and authors' reply. The authors' replies are shown in bold italics.
I would like to thank you very much for the opportunity to review the article „Experimental Investigation of Erosion Characteristics of Cohesive Sediments.” I find the article interesting, the subject matter is essential from the point of view of erosion of the bed or river banks, transport of river debris and silting of water reservoirs or backfilling of the river from its mouth.
Response: We thank the reviewer for providing the review of our draft submission. We want to emphasize that our submission is under the category of a "short communication" and not under the category of "articles".
However, before publication, the article requires a thorough rewriting. Needs additions of omitted descriptions of the material, the test methodology, an indication of a clear research objective, a sign of the originality of the article, and evidence of what new it brings to science.
Response: We appreciate the reviewer's comments. We have rewritten the draft m/s to a large extent taking into consideration all the comments provided by this reviewer. This has certainly enhanced the scientific quality of the m/s. The changes include those made in sections: Abstract, Introduction, Material and Methods, Results and Discussion and Conclusions.
The title of the article states, 'Cohesive Sediments.' Please explain what speaks for using the term "cohesive." From the point of view of grain size, it is non-cohesive material. In the manuscript, there is no description of this material to confirm this property, perhaps beyond the coagulation (flocculation) property given in the text. It would be appropriate to use the term "fine-grained material" - fine sediment, also used in the text.
Response: We have changed the original title to include the term " fine-grained". We agree that this provides clarification to readers.
Throughout the article, we find many references to literature. There are nine items, including seven by one author (Krishnappan, B. G., items 2, 9, 11, 15, 16, 23, 26) and two by another author (Collins, A.L., items 18, 19). Seven authors write the article. Therefore, please specify the role and percentage of each author in the report. This is a current practice in many journals.
Response: Every author of this m/s has contributed significantly to the study described in the paper. We are not sure where to specify the role and the percentage of each author's contribution in the submission since the journal system appears not to request this. We will provide this information if needed. The paper represents a product of international collaboration between Canada and UK.
So many references to literature related to the content of the article raise controversy as to whether the material is original and what brings new to science. Please address my concerns.
Response: We believe that the revised m/s on the whole addresses the reviewer's concern here. The development of experimental evidence for the upper River Taw using the methods described in the paper is novel for UK rivers and it is important to note that this type of evidence requires catchment-specific experimentation-there are no generic fixes and cohesive sediment models therefore require bespoke data on a study by study basis.
Notes to the Abstract. Too general and contains known knowledge of factors influencing erosion process in river beds. Bed shear stress and consolidation, however, are not the only factors, but I understand that boundary conditions to the model need to be generalized. It is also unknown what kind of debris is being considered. They are all alluvial deposits — rolling, sliding, and skipping. There is nothing mentioned about that. There is a lack of knowledge about the particles, granulometry, organic matter content, chemical composition of the material, the influence of agricultural management of adjacent areas, etc. Lack of information about the innovation of research, modeling, and results obtained.
Response: We have rewritten the abstract in the revised m/s. It is more specific and it provides the context for the study, approach used and outlines some of the significant results from the study-see lines 22-36 in the revised m/s.
Notes to the Introduction. It contains only a literature review, foreign and own. It follows that the article includes only repeated research and reference to existing theories and models. It consists of a description of the catchment area, land use adjacent to the river, geomorphology. But it is not then analyzed anywhere, e.g., in case of coagulation of silt particles.
Response: We have rewritten the introduction in the revised m/s. In the revised introduction we have added a figure showing the map of the study catchment and sampling locations. We have highlighted the need for the study, described the study area and outlined the rationale for the study more clearly- see lines 41-99 in the revised m/s.
Comments on the chapter Materials and Methods. Too general and referring directly to the literature. The chapter needs to be supplemented with a diagram of the test, a photo of the test stand, a way of mapping the in-situ state of the material from the river bed. The reader, and especially the reviewer, does not evaluate the cited literature but this particular article. Usually, the research is performed on many models and repetitions. Why cohesive samples, lack of explanations. There is one study described in the article, is it? Why? The shear stresses used in the test are in the order of 9E-2 Pa. Is it measured with such accuracy or recalculated? Why is this test methodology used? The applied mathematical model (Flume Model) is also used behind the literature (Krone). What's your own?
Response: We have modified the Material and Methods section in the revised m/s. All of the points raised by the reviewer with respect to this section are addressed in the revision. The Rotating annular flume in the study is fully described including the evaluation of the bed shear stress in the flume. The experimental procedure is elaborated. The methodology of Krone [1] is summarized, and the mathematical model of cohesive sediment transport in the rotating annular flume of Krishnappan [2] (FLUME model) is outlined - see lines 102-284 in the revised m/s.
The settling stage and the Flocculation stage were adapted from the cited literature. Where is the originality of the article?
Response: The originality of the model lies in the fact that the settling stage of the FLUME model was formulated using an erosion rate function that was derived from the approach of Krone [1]. This is the first time that such an erosion rate function has been used in the FLUME model.
Notes to the Results and discussion chapter. No presentation of the grain-size curve of the test material. Figure 1 presents only the distribution of the size of individual fractions and their percentage content by volume. This is an interesting presentation in relation to grain size changes before and after the test. In conclusion, the material has been classified as cohesive.
Response: The Results and Discussion section of the manuscript was also revised. The original Figure 1 was replaced with Figures 3 and 4. Figure 3 gives the size distribution of the eroded sediment samples undisturbed and sonicated. Figure 4 gives a photomicrograph of the eroded sediment in suspension. These two new figures show that the eroded sediment participates in the flocculation process suggesting that the sediment under investigation is indeed cohesive.
Please give the grain-size curve of the tested material and define it with the indicators based on grain size known from the literature. This will allow us to evaluate the material as one fractional or multi-fractional, show its resistance to filtration phenomena such as suffusion, etc. What does consolidation mean in this case? Consolidation is a process related to the load, the formation of excess water pressure in pores, and its dispersion. It is a process that occurs in a long time. It depends on the grain size of the particles, and the drainage conditions of the loaded layer, one or two vertical directions. What kind of settlement: immediate in undrained conditions, consolidation faze or secondary. What type of sedimentation: immediate, consolidation phase connected with the dissipation of excess pore water pressure or secondary in effective stress. How is it in this case?
Response: The grain size distribution of the eroded sediment is given in Figure 3 in the revised m/s. A discussion of the distribution is given on lines from 302 to 320. We did not examine the process of consolidation in any great detail because the thickness of the sediment bed tested were very thin (only a few mm). Therefore, the self weight consolidation that is active in thick sediment beds may not be significant in the present study. So, we have replaced the term "consolidation" with the term "age of deposit" in the revised m/s as detailed above.
What kind of test was performed to determine the grain size distribution? What is the procedure and type of device used? The bed shear stress values used in the analysis are very low. The testing time is short. What does this result from?
Response: The grain size distribution presented in Figure 3 in the revised m/s was measured using a LISST. The details are given on line 127-128 and 310-320. The bed shear stress values generated in the flume are within the range of the critical shear stresses for erosion and deposition of cohesive sediments and are commensurate with the properties of sediment used in the present study. The time of tests reflects the time scale of the cohesive sediment transport processes in rotating annular flumes.
The values given in figure 3 do not correlate with those given in table 1. The signature under the figure also. Why? What the authors wanted to show in the drawing. Please explain.
Response: Figure 3 of the original m/s was to plot the concentration of eroded sediment as a function of time for the erosion experiments and not for plotting the values shown in Table 1. The values in Table 1 are plotted in Figure 5 of the original m/s.
Notes to the chapter Conclusion. I don't know what to write. It's very general now. The chapter can be rewritten after all comments and corrections have been made.
Response: The conclusion section has been rewritten in the revised m/s and it summarizes the findings of this study - see lines 381-404 in the revised m/s.
Round 2
Reviewer 2 Report
Thanks to the authors for their answers and for many of mycomments. Not all of them have been included in the responses,
but I consider them to be sufficient.
Taking the explanations into account, I am currently assessing
the article positively.